# Hypoxia and hypoxia inducible factor-1α are required for normal endometrial repair during menstruation

Jacqueline A. Maybin [1], Alison A. Murray[1], Philippa T.K. Saunders [2], Nikhil Hirani[2], Peter Carmeliet[3] & Hilary O.D. Critchley [1]

Heavy menstrual bleeding (HMB) is common and debilitating, and often requires surgery due to hormonal side effects from medical therapies. Here we show that transient, physiological hypoxia occurs in the menstrual endometrium to stabilise hypoxia inducible factor 1 (HIF-1) and drive repair of the denuded surface. We report that women with HMB have decreased endometrial HIF-1α during menstruation and prolonged menstrual bleeding. In a mouse model of simulated menses, physiological endometrial hypoxia occurs during bleeding. Maintenance of mice under hyperoxia during menses decreases HIF-1α induction and delays endometrial repair. The same effects are observed upon genetic or pharmacological reduction of endometrial HIF-1α. Conversely, artificial induction of hypoxia by pharmacological stabilisation of HIF-1α rescues the delayed endometrial repair in hypoxia-deficient mice. These data reveal a role for HIF-1 in the endometrium and suggest its pharmacological stabilisation during menses offers an effective, non-hormonal treatment for women with HMB.

[1] MRC Centre for Reproductive Health, The Queen's Medical Research Centre, The University of Edinburgh, 47 Little France Crescent, Edinburgh, EH16 4TJ, Scotland. [2] MRC Centre for Inflammation Research, The Queen's Medical Research Centre, The University of Edinburgh, 47 Little France Crescent, Edinburgh EH16 4TJ, Scotland. [3] Laboratory of Angiogenesis and Vascular Metabolism, Vesalius Research Centre, Centre for Cancer Biology, KU Leuven, 3000 Leuven, Belgium. Correspondence and requests for materials should be addressed to H.O.D.C. (email: hilary.critchley@ed.ac.uk)

Heavy menstrual bleeding (HMB) is one of the most common reasons for attendance at gynaecology clinics, affecting 20–30% of pre-menopausal women[1]. Over 800,000 women per year seek treatment in the UK alone[1, 2]. Many women with HMB become anaemic and have significantly decreased quality of life[3]. A US study highlighted the substantial socioeconomic impact of HMB, with financial losses of greater than $2000 per patient per year as a result of work absence and additional home management costs[4]. Medical treatments for HMB are available but are often intolerable for women due to lack of efficacy or troublesome hormonal side effects, including prevention of fertility, bloating, unscheduled bleeding, fatigue and depression. A national 4-year audit of HMB in the UK found 43% of women received surgery within a year of their first attendance at hospital[1]. Considering 47% of all UK-born babies are to women aged ≥30[5], fertility-ending surgery is frequently unacceptable. In addition, surgery introduces risk of organ damage, haemorrhage and infection. There is a clear unmet clinical need for non-hormonal, fertility-preserving and cost-effective medical treatments for this debilitating condition.

In the absence of pregnancy, a sharp decline in ovarian progesterone (P4) production occurs as the corpus luteum regresses. This triggers an inflammatory response in the local endometrial environment, culminating in tissue destruction, shedding of the luminal two thirds of the endometrium (the functional layer) and menstrual bleeding (Fig. 1a). Cessation of menses requires (i) vasoconstriction of specialised endometrial spiral arterioles, (ii) an effective haemostatic response, including repair of damaged vasculature and (iii) timely reepithelialisation of the remaining denuded basal endometrium[6, 7]. This unique, scarless repair process is essential to maintain fertility and limit menstrual blood loss (MBL) and has immense translational potential. To date, the mechanisms and regulation of endometrial repair are not well understood.

The possibility that hypoxia has a physiological role at menses was proposed over 70 years ago[8]. Direct observation of endometrial explants transplanted to the eye of Rhesus macaques revealed intense vasoconstriction of spiral arterioles and focal bleeding following P4 withdrawal. Since this publication, the presence and role of hypoxia in the endometrium has remained the subject of intense debate, with conflicting results from different laboratories utilising in vitro, ex vivo and in vivo models[9–13].

Hypoxia inducible factor (HIF) is the master regulator of the cellular response to hypoxia, having well defined roles in angiogenesis, mitogenesis, apoptosis, inflammation and metabolism at other tissue sites[14–16]. When oxygen is abundant, the alpha subunit (HIF-1α) is hydroxylated by prolyl hydroxylase (PHD) enzymes which triggers its rapid degradation by the proteasome[17]. In hypoxia, the oxygen-dependent PHD enzymes are inactive and HIF-1α protein is stable. HIF-1α can then bind to the beta subunit (HIF-1β) and induce the transcription of downstream targets that promote adaptation to a hypoxic environment[18]. HIF-2α is an alternative binding partner for HIF-1β and appears to have overlapping but distinct target genes compared to HIF-1α[19]. The role of hypoxia and HIF in human endometrial function remains undetermined.

In studying the role of hypoxia and HIF-1 at menstruation using human endometrial tissue and mouse models of simulated menses, we reveal hypoxia stabilises HIF-1α in the menstrual endometrium to enable timely repair of the denuded endometrial surface. In addition, we highlight a potential use for PHD inhibitors (HIF-1α stabilisers) as a transient, non-hormonal treatment for women with HMB.

## Results

### Menstrual endometrial HIF-1α is decreased in women with HMB. HIF-1α mediates the cellular response to hypoxia and thus

is a marker of tissue hypoxia. To investigate whether levels of hypoxia/HIF-1α differ across the menstrual cycle, we extracted nuclear protein from well characterised endometrial biopsies from women with subjectively normal menstrual bleeding (NMB) at various stages of the menstrual cycle (Supplementary Table 1). HIF-1α protein was detected in endometrial tissue from the late secretory (LS), menstrual (M) and one sample from the early proliferative phase (P) of the menstrual cycle (Fig. 1b), i.e. stages of the menstrual cycle when progesterone levels are low (Fig. 1a). HIF-1α was not detected during the later proliferative, early- or mid-secretory phases (ES, MS). Densitometry revealed HIF-1α levels were highest during the menstrual phase (Fig. 1b). HIF-2α, an alternative binding partner for HIF-1β, was not detected in human endometrial biopsies taken during the peri-menstrual phase but was detected during the early-mid secretory phase (Fig. 1c).

After determining that HIF-1α, but not HIF-2α, was present at a cycle stage consistent with active menstruation and repair, we investigated the presence of HIF-1α during menstruation in women with objectively measured MBL (Supplementary Table 1) to determine if an aberrant endometrial hypoxia/HIF-1 response is associated with HMB (blood loss > 80 ml/cycle). Western blots of whole endometrial tissue nuclear extracts revealed that women with HMB had significantly lower levels of HIF-1α at menstruation than those with NMB (Fig. 1d). There were no significant differences in endometrial HIF1A or HIF2A between women with NMB compared to HMB at any cycle stage (Supplementary Fig. 1). As there were no differences in HIF-1α transcript during menstruation in women with HMB versus normal loss, we concluded that endometrial HIF-1α is post-transcriptionally regulated, most likely due to protein stabilisation.

Stabilisation of HIF-1α during normal menstruation would be expected to lead to an increase in downstream HIF-1 targets. We examined VEGF and CXCR4 as known targets of HIF-1 that have putative roles in endometrial repair and regeneration[12, 20–22]. These HIF-1 targets were dramatically increased in endometrial samples collected during the menstrual phase from women with NMB (Fig. 1e), consistent with high levels of HIF-1 at this stage of the cycle. In contrast, women with HMB showed no increase in VEGF levels during the menstrual phase. The twofold increase in endometrial CXCR4 mRNA concentration during menstruation in women with NMB was similarly absent in women with HMB.

As HIF-1 is known to accelerate wound repair via stimulation of mitogenesis and angiogenesis at other tissue sites[23, 24], we aimed to determine the functional impact of decreased HIF-1α and its downstream targets in the endometrium at menstruation. We measured duration of bleeding as a clinical marker of endometrial repair. Women with HMB were found to bleed for 2 days longer on average than women with NMB, consistent with defective repair (Fig. 1f).

### Hypoxia is present in a mouse model of simulated menses. Our observational human data are consistent with a defective hypoxic response in the endometrium at menstruation. However, to detect and manipulate endometrial hypoxia we utilised a mouse model of endometrial breakdown and repair that simulates menstruation. Ovariectomised mice were treated sequentially with E2 and P4 to replicate the proliferative and secretory phases of the human cycle. Decidualisation of the endometrium is required for active menses and occurs spontaneously in humans during the secretory phase. This was replicated in our model by a transcervical injection of oil, resulting in a decidualisation reaction. Subsequent withdrawal of P4 triggered menstrual-like bleeding 8 h later (T8 = active bleeding, 'menstruating') (Fig. 2a). By 24 h

following P4 withdrawal endometrial reepithelialisation and stromal restoration was underway (T24 = endometrial repair).

Pimonidazole is a marker of tissue hypoxia, with positive staining only detected in tissues where oxygenation is <10 mmHg. In our simulated menses model there were obvious temporal changes in pimonidazole staining following P4

withdrawal (Fig. 2b). Hypoxia was not detected prior to menstrual bleeding (T0, when P4 levels were high). During simulated menstruation (T8, 8 h after P4 withdrawal), pimonidazole staining was intense and localised to the area of endometrial separation, i.e. the area requiring repair. At the time of endometrial repair (T24, 24 h after P4 withdrawal)

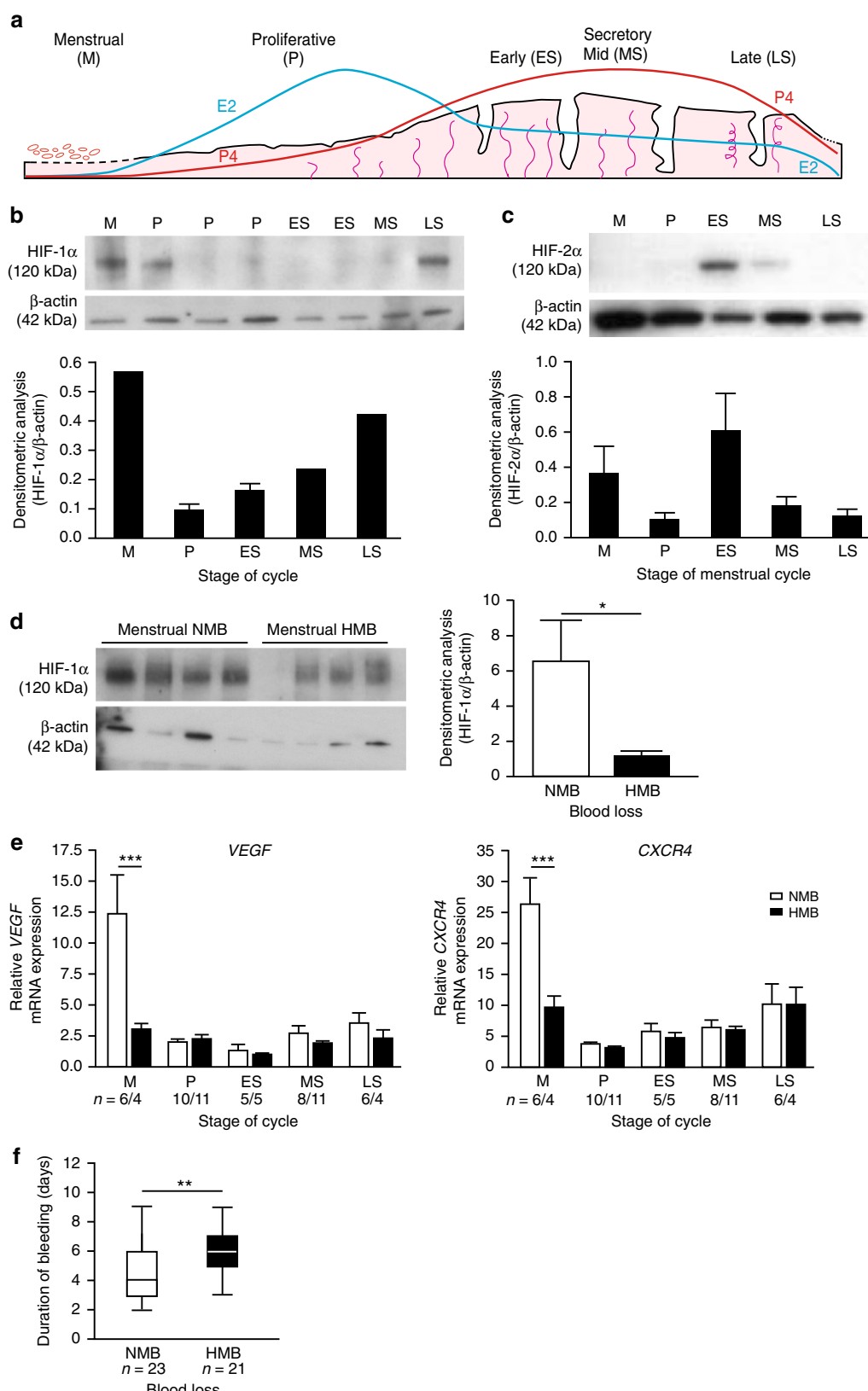

pimonidazole staining was greatly reduced; where present it was confined to non-reepithelialised epithelium that had not yet fully repaired. Thus a transient hypoxic episode was detected in the endometrium during normal menstruation and was localised to the areas requiring repair. We postulated that this local hypoxia regulated endometrial HIF-1α levels during menses.

**Lack of hypoxia reduces HIF-1α and prevents efficient repair.** To test the hypothesis that local hypoxia regulates endometrial HIF-1α, we subjected C57BL/6 mice to hyperoxia (75% O2) at the time of P4-withdrawal (T0) and compared them to mice maintained in normoxia (21% oxygen). Hyperoxia prevented local endometrial hypoxia at menses, evidenced by minimal pimonidazole staining 8 h after imposition of hyperoxia concurrent with P4 withdrawal (Fig. 3a, upper panel). In contrast, strong pimonidazole staining was detected in the endometrium of mice maintained in normoxia (Fig. 3a, lower panel), indicating physiological hypoxia at simulated menses. Consistent with hypoxia being the trigger for HIF-1α induction at menses, HIF-1α protein levels were significantly reduced 8 h following P4 withdrawal (T8) in the endometrium of mice under hyperoxia versus those in normoxia (Fig. 3b).

Histological scoring of uterine sections was used to quantify endometrial breakdown and repair from 1 (decidualisation) through to 5 (full repair) (Supplementary Fig. 2). Assessment of endometrial breakdown/repair at T8 revealed no significant differences between mice under hyperoxia versus normoxia (Fig. 3c), indicating no difference in endometrial breakdown/onset of menses. Examination of tissue at the time of expected repair (T24) revealed 56% of mice in normoxia (hypoxic menstruation) reached complete repair, compared to only 22% of mice where hypoxia was prevented during menses (hyperoxia), although median repair scores were not significantly different between groups (Fig. 3d).

A panel of known HIF-1 targets and inflammatory mediators were examined by PCR in uterine tissue from mice in these experimental groups (normoxia/hyperoxia) at T8 and T24 (Fig. 3e, Supplementary Fig. 3). HIF-1 regulated mRNAs were increased at T8, corresponding with physiological endometrial hypoxia. In contrast, inflammatory mRNAs were increased at T24, subsequent to the transient hypoxic episode. Prevention of menstrual hypoxia had no significant impact on *Vegf* or *Cxcr4* mRNA concentrations. The pattern of *Cxcr4* was more consistent with other inflammatory mediators investigated (Supplementary Fig. 3), displaying an increase at T24 rather than T8, i.e. after the endometrial hypoxic response. In contrast, *Vegf* was maximal at T8 and showed a non-significant decrease in hyperoxic mice versus normoxic conditions, suggesting hypoxic/HIF-1 regulation (Fig. 3e). This non-significant trend was consistent in all HIF-1 targets investigated; *Adm*, *Glut-1* and *Ldha* (Supplementary Fig. 3), consistent with a sub-optimal HIF-1 response in hypoxia-deficient menses. We conclude that hypoxia is required to stabilise HIF-1α in the menstrual endometrium to drive optimal production of downstream repair factors.

**Hif-1α^{+/−} mice have delayed endometrial repair.** To confirm the role of HIF-1α in the menstrual endometrium, we compared endometrial breakdown/repair in wild type (*Hif-1α^{+/+}*) and HIF heterozygote (*Hif1a^{+/−}*) mice. HIF-1α^{−/−} knockout mice are embryonically lethal due to cardiovascular and neural tube defects[25]. In contrast, HIF-1α^{+/−} mice are phenotypically normal but lack the ability to mount an appropriate HIF-1 response to hypoxic conditions (i.e. at menstruation)[26, 27], a scenario we postulate is analogous to the endometrium of women with HMB at menses (Fig. 1d, Supplementary Fig. 4a). Importantly, HIF-2α did not increase during bleeding to compensate for HIF-1α deficiency (Supplementary Fig. 4b). When actively bleeding at T8, there was no significant difference in histological endometrial breakdown/repair score, suggestive of no difference in onset of menses (Fig. 4a). In contrast, during the repair phase at T24, the score was significantly lower in *Hif1a^{+/−}* mice versus controls (Fig. 4b), consistent with delayed endometrial repair. No significant differences in *Vegf* or *Cxcr4* were observed between HIF-1α heterozygotes and wild types at any time-point (Fig. 4c). *Ldha* was significantly decreased in HIF-1α heterozygous mouse uterine tissue at T8 (Supplementary Fig. 4c). As carbonic anhydrase IX (CaIX) is one of the most sensitive endogenous sensors of HIF-1 activity[28], we examined its presence by immunohistochemistry in *Hif1a^{+/−}* versus wild-type mice at T8 (time of active bleeding and maximal hypoxia). Similar to pimonidazole, CaIX staining was localised to the area of endometrial separation (Fig. 4d). CaIX staining was reduced in *Hif1a^{+/−}* mice, consistent with a reduced HIF-1 response in these HIF-deficient mice. We conclude that HIF-1α deficiency at menstruation results in significantly delayed endometrial repair.

**Pharmacological inhibition of HIF-1 at menses delays repair.** To provide acute pharmacological manipulation of Hif-1α at menses, C57BL/6J mice were treated with echinomycin (1 mg kg^{−1}), which inhibits HIF-1 binding to hypoxia-response elements on target genes[29]. Conversely, 'pseudohypoxia' was induced using DMOG (8 mg), a PHD inhibitor that stabilises HIF-1α even in normoxia[30]. Drugs were administered 24 h prior to and at the time of P4-withdrawal (T0) and histological breakdown/repair quantified at T8 (active bleeding, 'menstruation') and T24 (during repair).

At the time of bleeding (T8), all treatment groups showed features of decidualisation or breakdown (Fig. 5a). Echinomycin treatment was associated with an earlier stage of breakdown when compared to vehicle but not DMOG-treated mice. More significantly, echinomycin treatment delayed endometrial repair at T24, showing little difference between tissues recovered at T8 and T24 (Fig. 5b). Uterine morphology in echinomycin-treated mice was characteristic of early stages of breakdown 24 h after P4 withdrawal, contrasting markedly with vehicle or DMOG-treated mice that had progressed into the repair phases. Interestingly, stabilisation of HIF1α with DMOG had no additional effect over vehicle treated animals (Fig. 5b). These data suggest the physiological hypoxic response at menses is sufficient for

**Fig. 1** HIF-1α is present in the human endometrium at menstruation and reduced in women with HMB. **a** The human menstrual cycle. E2 estradiol, P4 progesterone. **b** Western blot and densitometry for HIF-1α in nuclear protein extracts from endometrial samples taken at different phases of the menstrual cycle. (*n* = 1–3 biological replicates per stage of cycle). **c** HIF-2α western blot of endometrial nuclear protein extracts from across the menstrual cycle (Representative image from *n* = 3 biological replicates per stage of cycle.). **d** HIF-1α protein blot and densitometry in menstrual phase endometrium from women with NMB (*n* = 4) and HMB (*n* = 4). *P < 0.05, unpaired *t*-test. **e** *VEGF* and *CXCR4* mRNA concentrations in endometrium from women with HMB and NMB at each stage of the menstrual cycle. ***P < 0.001, two-way ANOVA with Bonferroni's multiple comparison's test. **f** Duration of bleeding in days for women with objectively measured NMB and HMB. **P < 0.01, unpaired *t*-test. Box and whisker plots: box represents upper and lower quartiles with horizontal line representing the median, whiskers represent minimum and maximum values. M menstrual, P proliferative, ES early secretory, MS mid secretory, LS late secretory. HMB=heavy menstrual bleeding, >80 ml; NMB=normal menstrual bleeding, <80 ml. Uncropped western blots included in Supplementary Fig. 8.

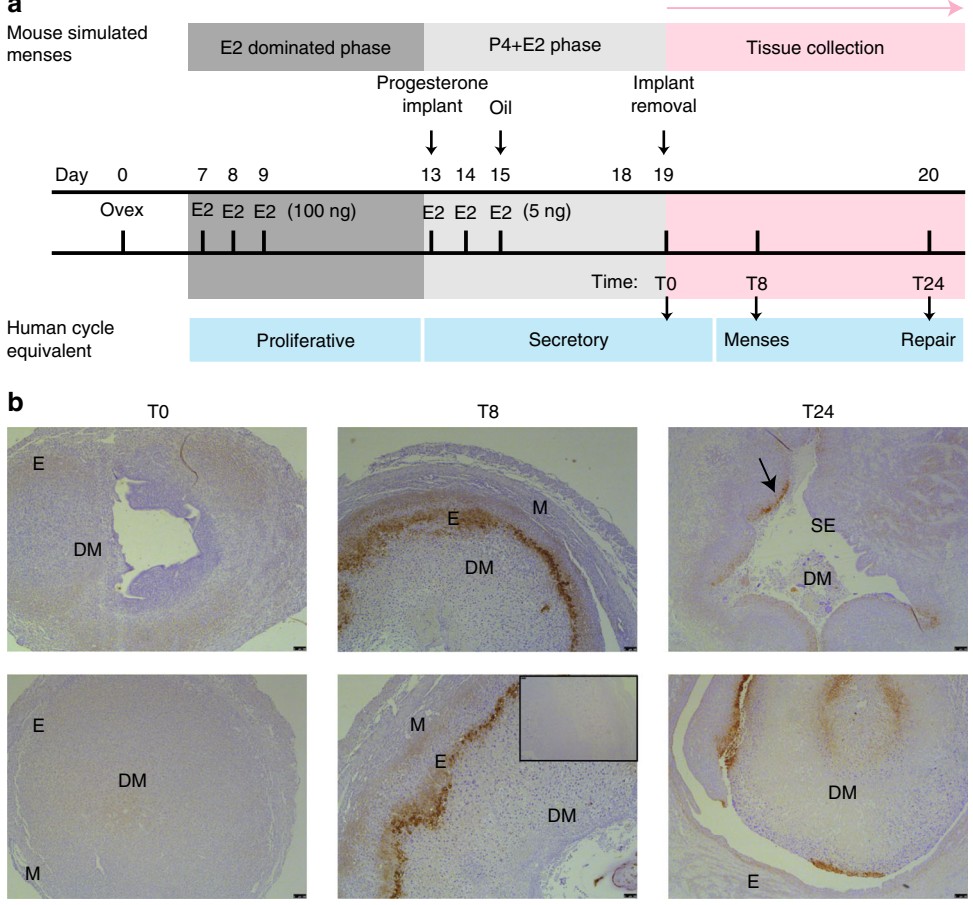

**Fig. 2** The mouse model of simulated menstruation reveals physiological hypoxia at menses. **a** Murine model of simulated menstruation. E2 estradiol, P$_4$ progesterone, Ovex ovariectomy. T0 time of P4 implant removal, T8 8 h following progesterone withdrawal, T24 24 h following P4 withdrawal. **b** Representative endometrial pimonidazole staining from two mice T0, T8 and T24 (inset = negative control). Total replicates T0 ($n = 4$), T8 ($n = 7$) and T24 ($n = 11$). DM decidualised mass, E endometrium, M myometrium, SE surface epithelium. Arrow indicates positive staining in endometrium that has not yet reepithelialised. Scale bar = 100 μm

maximal HIF-1α stabilisation and confirms that HIF-1 is essential for normal endometrial repair at this time.

Downstream targets of HIF-1 were examined in mice treated with echinomycin, vehicle and DMOG at T0 (prior to P4 withdrawal), T8 (during bleeding) and T24 (repair phase). In vehicle controls, *Vegf* mRNA was again maximal at T8 (Fig. 5c). Echinomycin treatment prevented this T8 increase in *Vegf*, consistent with a defective HIF-1 response (Fig. 5c). Echinomycin treatment also significantly decreased uterine *Cxcr4* at T24 versus DMOG treatment but not vehicle. Again HIF-1 target genes displayed similar patterns to *Vegf* and inflammatory mediators were comparable to *Cxcr4* (Supplementary Fig. 5a). Consistent with these chemokine data, neutrophils were increased in uterine tissue at T24 and there was a marked decrease in tissues from mice treated with echinomycin (Fig. 5d). There were no detectable differences with HIF-1 pharmacological inhibition/ stabilisation in neutrophil numbers at T8 or macrophage numbers at either timepoint (Supplementary Fig. 5b).

As a reduction in HIF-1 at menses decreased *Vegf* and other HIF-1 targets, we examined the functional consequences of these aberrations during endometrial repair by examining surface epithelial cell proliferation and endometrial vasculature in our experimental groups. Immunohistochemical staining of bromo-deoxyuridine (BrdU) incorporation (Fig. 5e) showed intense staining of luminal epithelium of vehicle treated mice, consistent with high rates of proliferation to reepithelialise the shed

endometrial surface. This was even more marked in DMOG-treated mice. In contrast, there was minimal incorporation of BrdU in the endometrium of echinomycin-treated mice suggesting that inhibition of HIF-1 prevented the normal proliferation of the surface epithelial cells and delayed reepithelialisation.

Uterine endothelial cells (CD31-positive cells) in echinomycin-treated mice had a disorganised appearance, without evidence of lumen formation or pericyte coverage (α smooth muscle actin (αSMA) staining) when compared to DMOG-treated animals (Fig. 5f). This is suggestive of a disordered angiogenic response at menstruation, which may have significant impact on vascular function and MBL in subsequent cycles.

**Human endometrial HIF-1α affects endothelial branching.** Alongside our mouse model, we wished to test if reduced HIF-1α had similar effects in human endothelial cells. To do this we silenced *HIF1A* in human endometrial epithelial cells incubated in hypoxia and confirmed *HIF1A* specific knockdown (Supplementary Fig. 6). Conditioned media from *HIF1A* silenced and control cells was collected and used to treat human umbilical vascular endothelial cells to examine impact on endothelial branch formation (Fig. 5g). Addition of media conditioned by untransfected epithelial cells grown under hypoxia to HUVEC cells significantly increased HUVEC branching over that from normoxic incubation. However, branching was significantly

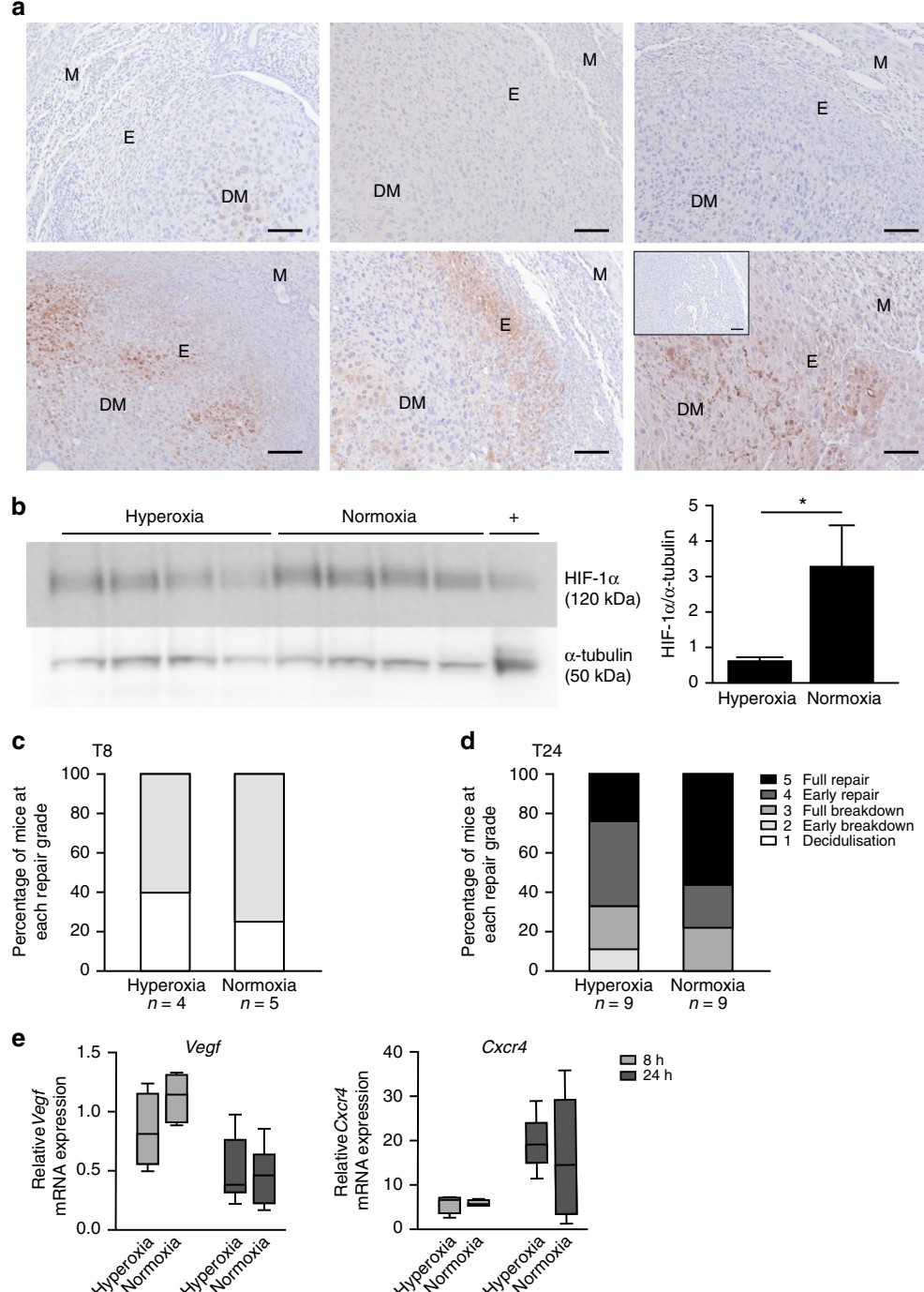

**Fig. 3** Non-hypoxic menstruation decreases endometrial HIF-1α and delays endometrial repair. **a** Pimonidazole staining in endometrium at T8 from three mice placed in hyperoxic (75% O2, top panel) and three mice in normoxic (21% O2, bottom panel) conditions at the time of progesterone withdrawal. M myometrium, DM decidualised mass, E endometrium. Inset = negative control. Scale bar = 100 μm. **b** HIF-1α western blot of uterine protein extracts at T8 from mice in 75% O2 (hyperoxia = non-hypoxic endometrium) versus 21% O2 (normoxia = hypoxic endometrium) with densitometry. *P < 0.05, unpaired t-test. **c** Endometrial histological breakdown/repair score at T8 in mice incubated in hyperoxic and normoxic conditions. Graphs represent percentage of mice at each histological grade per experimental group. White—decidualisation, light grey—early breakdown, medium grey—full breakdown, dark grey— early repair, black—full repair (Supplementary Fig. 2). **d** Endometrial histological breakdown/repair score at T24 in mice placed in hyperoxic and normoxic conditions. Graphs represent percentage of mice at each histological grade per experimental group. **e** Vegf and Cxcr4 concentrations in mouse uterine tissue at T8 and T24 in mice incubated in normoxia and hyperoxia. Box and whisker plots: box represents upper and lower quartiles with horizontal line representing the median, whiskers represent minimum and maximum values

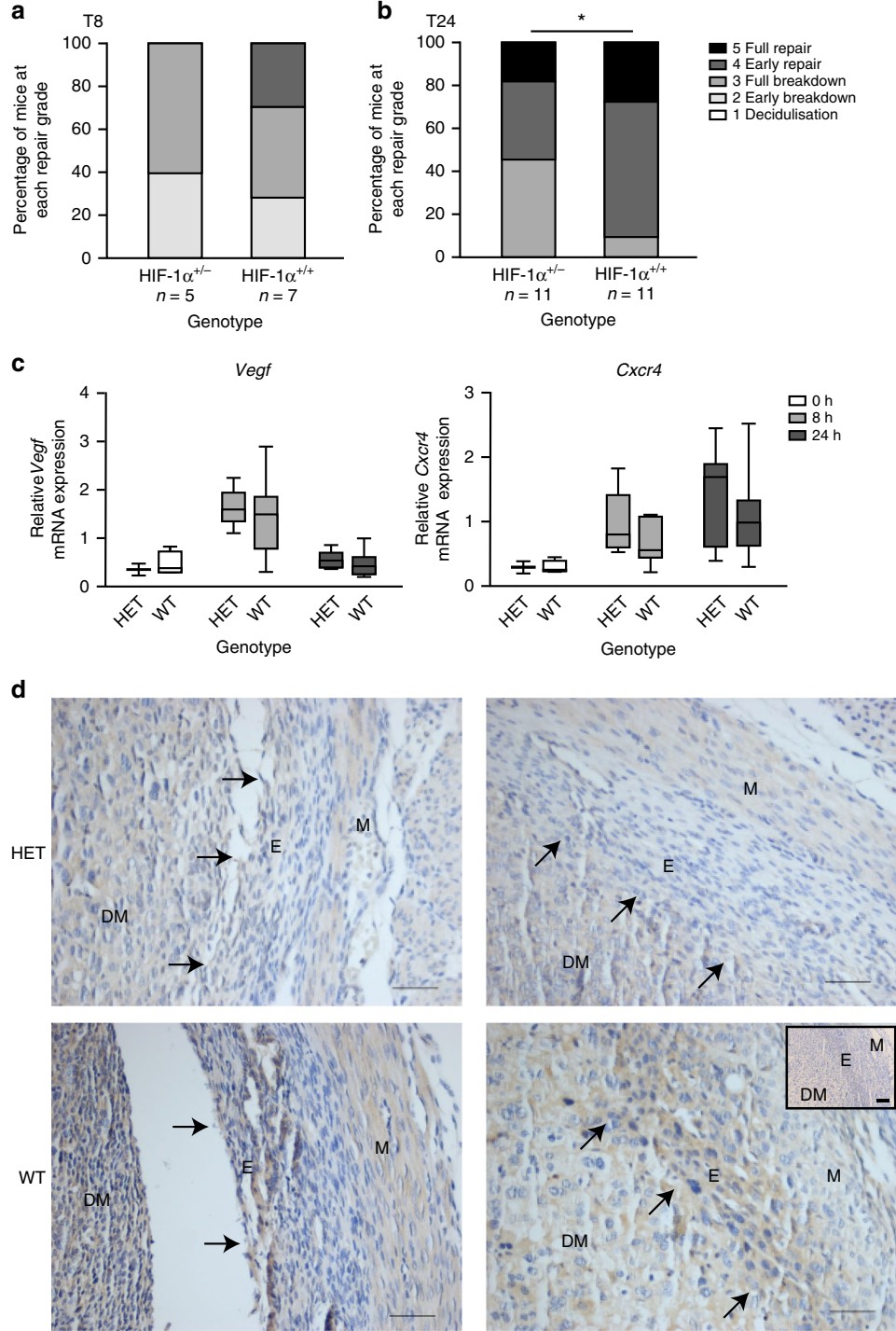

**Fig. 4** HIF-1α heterozygote mice display delayed endometrial repair following progesterone withdrawal. **a** Endometrial histological breakdown/repair score at 8 h following P withdrawal (T8) and **b** at 24 h following P withdrawal (T24) in HIF-1α heterozygote (HET, HIF-1α$^{+/-}$) versus wild type (WT, HIF-1α$^{+/+}$) mice. Graphs represent percentage of mice at each histological grade per experimental group. *$P < 0.05$, unpaired $t$-test. **c** Relative mRNA concentration of *Vegf* and *Cxcr4* in uterine tissue from HET versus WT mice at T0 ($n = 2/4$), T8 ($n = 5/7$) and T24 ($n = 11/11$). Box represents upper and lower quartiles with horizontal line representing the median, whiskers represent minimum and maximum values. **d** Carbonic anhydrase IX immunohistochemical staining in HIF-1α heterozygote (HET) and wild-type (WT) mice. Arrows represent area of endometrial separation. M myometrium, E endometrium, DM decidualised mass. Inset = negative control. Scale bar = 100 μm

attenuated if HUVEC cells were treated with medium conditioned by epithelial cells grown in hypoxia but in which shRNA had been used to knock down *HIF1A*. Two different ShRNA constructs had the same effect (1470/2192). We have previously

demonstrated that endometrial cells grown under hypoxia release VEGF[12]. To test whether VEGF could restore the endothelial branching potential of supernatant from *HIF1A* silenced cells, we added back rhVEGF, and found HUVEC branching was rescued.

**DMOG restores endometrial repair in non-hypoxic menses.** Our hyperoxic mouse model of menstruation demonstrated that local hypoxia prevention decreased endometrial HIF-1α and resulted in suboptimal endometrial repair, analogous with our findings in the endometrium of women with HMB. Therefore, we

aimed to improve endometrial repair by pharmacologically increasing HIF-1α at menses. DMOG stabilises HIF-1α, even in normoxic conditions, by inhibiting PHD enzymes involved in HIF-1α breakdown, i.e. induces 'pseudohypoxia'. C57BL/6J mice were treated with vehicle or DMOG 24 h prior to P4 withdrawal

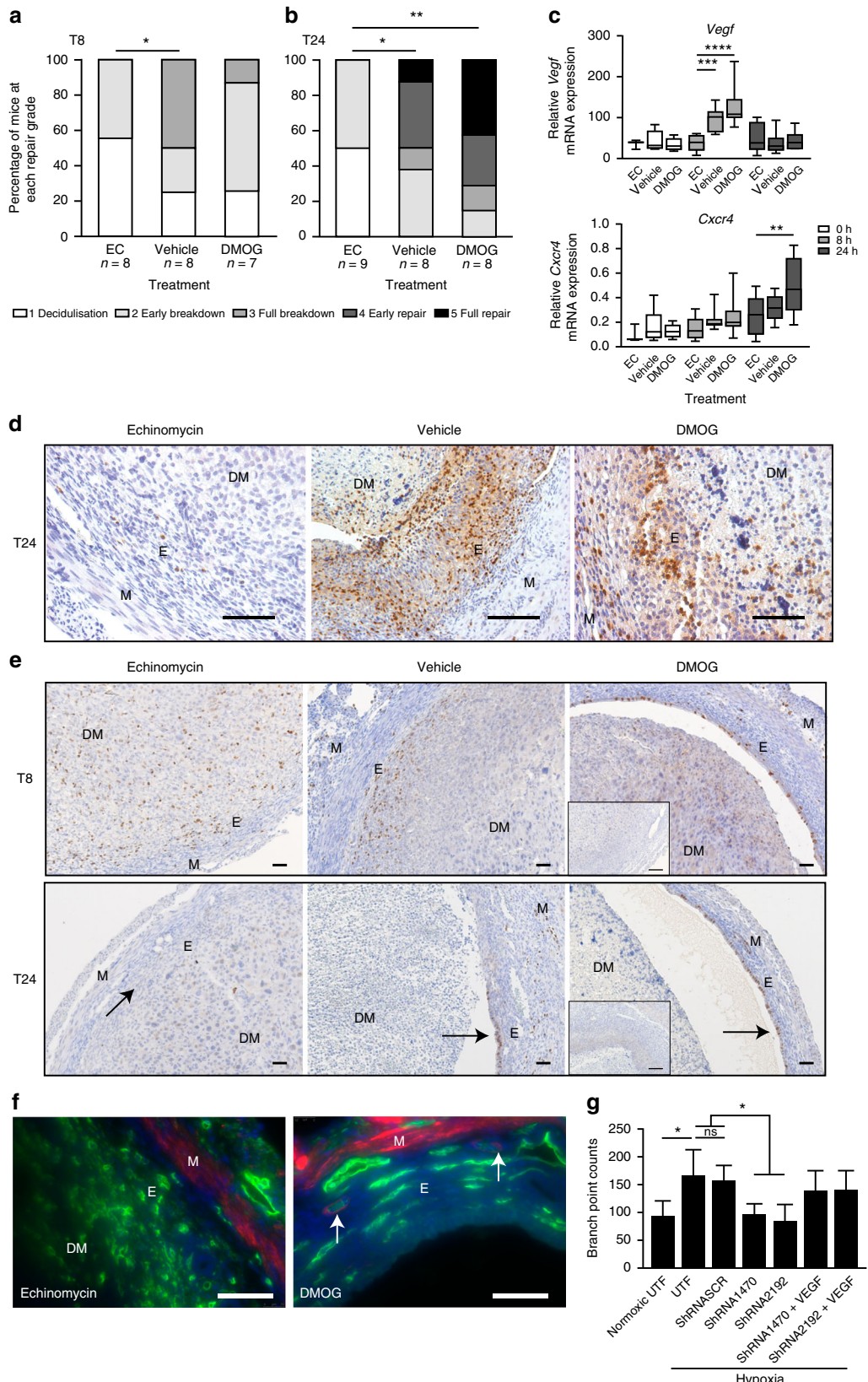

and at the time of P4 withdrawal (T0) and all mice were placed in hyperoxia at T0 to prevent endometrial hypoxia at menstruation (Fig. 6a). At 24 h following P4 withdrawal (T24), mice treated with DMOG had significantly greater histological repair score than mice treated with vehicle (Fig. 6b), providing evidence that DMOG can rescue the delayed endometrial repair observed when physiological hypoxia is not present at menstruation.

## Discussion

Excessive menstruation is common and debilitating, resulting in significantly decreased quality of life for women and a major socio-economic burden. Current medical treatments often have unacceptable hormonal side effects or are ineffective[31]. We reveal that women with HMB have reduced endometrial HIF-1α and downstream targets and prolonged menstrual bleeding, suggestive of delayed endometrial repair. Herein, we detected hypoxia during active bleeding in the mouse model of simulated menses. Prevention of menstrual hypoxia decreased HIF-1α and delayed repair of the denuded endometrium. Deficient endometrial HIF-1α following P4-withdrawal in our mouse model was associated with delayed endometrial repair. Finally, we revealed that administration of DMOG (HIF stabiliser) at the time of simulated menstruation in our model of non-hypoxic menstruation significantly improved endometrial repair. These results confirm that pharmacological stabilisation of HIF-1α at menstruation has the potential to fulfil the unmet clinical need for a novel, non-hormonal therapeutic strategy for women with HMB.

As HIF has two transcriptionally active forms with distinct but overlapping target genes[18, 19], we examined both subunits across the menstrual cycle. We found endometrial HIF-1α was present exclusively during the peri-menstrual phase (luteo-follicular transition), when vasoconstriction of spiral arterioles and local endometrial hypoxia are expected to occur[8]. Endometrial HIF-2α was not present during menstruation, but only during the early to mid-secretory phase. This temporal pattern suggests HIF-2α does not have a key role in the regulation of menstruation, though it remains possible it may contribute to endometrial function at other stages. This is consistent with in vitro studies of breast cancer cells, where HIF-2α contributed to few, if any, of the transcriptional responses to acute hypoxia[18]. We also revealed that women with objectively defined HMB (>80 ml per cycle) had decreased endometrial HIF-1α protein and its downstream targets VEGF and CXCR4 during menstruation when compared to women with NMB. These carefully categorised tissues with linked objectively measured MBL suggest a role for hypoxia and HIF-1 in menstrual physiology and that aberrations may lead to HMB. Therefore, we utilised our mouse model of simulated menstruation to delineate the role of hypoxia and HIF-1α at menstruation.

Herein we detected a transient but intense hypoxic episode in the mouse endometrium during active bleeding, localised to the area in need of repair. This is in direct conflict with the xenograft menses model, where explants of human endometrial tissue were grafted subcutaneously and E2 and P4 pellets inserted and withdrawn after 60–90 days to induce menstruation[9]. Hypoxia was not detected using HIF-1α or pimonidazole immunohistochemistry. These differences may be explained by disturbance of the full thickness endometrial architecture in the immunodeficient xenograft model, where spiral arteriole function and immune cell function will be drastically modified. Our results are consistent with findings in the mouse model of simulated menstruation from within and outwith our department[11, 32].

A number of publications describe non-hypoxic regulation of HIF-1α with increases in HIF-1α protein in normoxic conditions by inflammatory mediators, such as prostaglandins and tumour necrosis factor α[33, 34]. As the local endometrial environment at menses has many features of an acute inflammatory response, with increases in prostaglandins, cytokines/chemokines and an influx of leukocytes[35], the contribution of hypoxia to HIF-1α stabilisation remained unclear. To address the role of hypoxia in HIF-1α regulation, we prevented menstrual hypoxia in our mouse model and observed a significant reduction in HIF-1α protein. Therefore, endometrial hypoxia is necessary for an appropriate HIF-1α response in vivo at menses. These findings are consistent with our previous human data, where hypoxic induction of VEGF and ADM in human endometrial cells was dependent on HIF-1α but that increases observed with normoxic prostaglandin E2 treatment were independent of HIF-1α[12, 36]. Hypoxic conditions were previously found to be unnecessary for MMP-induced menstrual breakdown in human cells and tissue explants cultured in vitro[10, 37]. Herein, our novel in vivo model confirmed that menstruation occurred despite the absence of hypoxia, but revealed that endometrial repair was delayed.

Genetic reduction of HIF-1α in our mouse model of simulated menses demonstrated normal onset of active bleeding but significantly delayed endometrial repair. Echinomycin treatment of mice to inhibit menstrual HIF-1 also significantly decreased HIF downstream targets and delayed endometrial repair. The decrease in HIF downstream targets was much less marked in HIF-1α heterozygotes, with no differences in Vegf and a non-significant trend towards reduction in Adm, Glut1 and Ldha in heterozygote mice versus wild type. This is not unexpected considering our model was a HIF-1α deficient, rather than a knockout animal and that the hypoxic episode was comparable between the two groups. Furthermore, localised endometrial hypoxia may result in localised, clinically significant alterations in HIF-1 targets, as demonstrated by changes in carbonic anhydrase protein. Localised changes in mRNA may not be detected in our analysis of

**Fig. 5** Pharmacological inhibition of HIF-1α delays endometrial repair in a mouse model of simulated menstruation. **a** Histological breakdown/repair score at 8 h after P4 withdrawal (T8) in wild-type mice treated with echinomycin (EC), vehicle or dimethyloxalylglycine (DMOG) and **b** 24 h following P4 withdrawal (T24). *P < 0.05, **P < 0.01, ANOVA with Dunn's multiple comparison's test. Graphs represent percentage of mice at each histological grade per experimental group. **c** Relative mRNA concentration of Vegf and Cxcr4 in uterine tissue at T8 and T24 from mice treated with echinomycin, vehicle or DMOG. Box represents upper and lower quartiles with horizontal line representing the median, whiskers represent minimum and maximum values. **P < 0.01 ***P < 0.001 ****P < 0.0001, two way ANOVA with Tukey's multiple comparison's test. **d** Ly6G staining in endometrium collected 24 h following progesterone withdrawal (T24) from mice treated with echinomycin (EC: HIF-1 inhibitor), vehicle or dimethyloxalylglycine (DMOG: HIF-1α stabiliser). M myometrium, DM decidualised mass, E endometrium. Scale bar = 100 μm. **e** BrdU staining in mouse uterine tissue at T8 and T24 in mice treated with echinomycin, vehicle or DMOG. Arrows in T24 represent areas of active proliferation. Inset = negative control, scale bar = 100 μm. **f** Endothelial cells (CD31, green) in echinomycin-treated animals at T24 display an abnormal, disorganised appearance with no detection of the pericyte marker αSMA (red), as seen in DMOG-treated mice (arrows). Nuclear staining (DAPI, blue). Representative images from n ≥ 6 per treatment group. Scale bar = 100 μm. **g** Human umbilical vein endothelial cell (HUVEC) branch point number in vitro after 8 h treatment with the conditioned media from (i) normoxic untransfected endometrial epithelial cells, (ii) hypoxic untransfected epithelial cells, (iii) hypoxic epithelial cells transfected with a scrambled control sequence. (iv, v) Hypoxic HIF-1α silenced epithelial cells or (vi, vii) HIF silenced cells in hypoxia with add back of rhVEGF (300 ng ml⁻¹). n = 3. UTF untransfected cells, SCR scrambled sequence negative control, 1470/2192: two different ShRNA constructs against HIF-1α. ns non-significant, *P < 0.05, ANOVA with Tukey's multiple comparison's test. Error bars = SEM

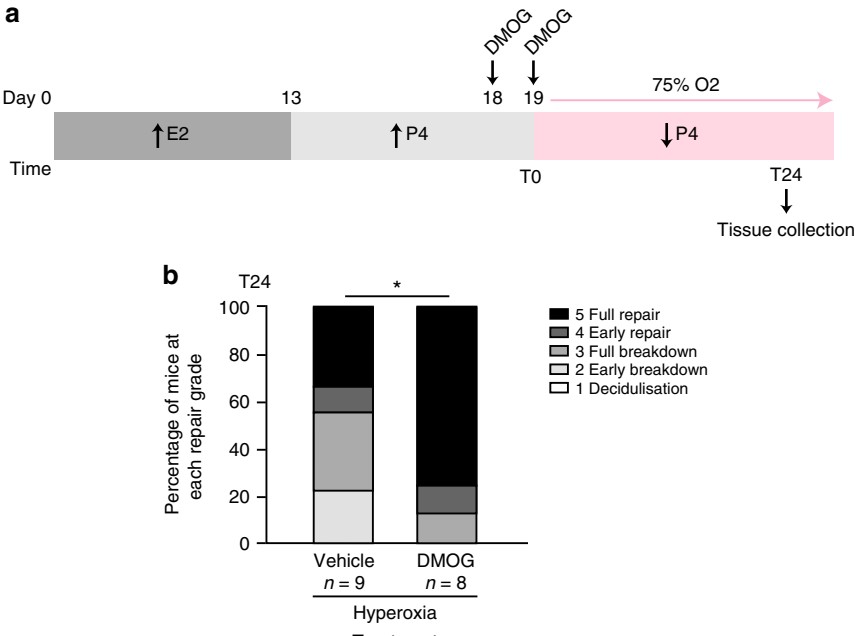

**Fig. 6** DMOG rescues the delayed endometrial repair phenotype in the mouse non-hypoxic menstruation model. **a** Mouse non-hypoxic menstruation model with intraperitoneal dimethyloxalylglycine (DMOG) treatment 24 h before and at the time of P4 withdrawal (T0). E2 estradiol, P4 progesterone. **b** Endometrial histological breakdown/repair score 24 h following P4 withdrawal (T24) for mice treated with DMOG or vehicle prior to placement in hyperoxic conditions (75% O2). Graphs represent percentage of mice at each histological grade per experimental group. *$P < 0.05$, Mann–Whitney test

whole uterine samples. Small changes in many HIF-1 downstream targets may explain the delayed endometrial repair phenotype observed.

Interestingly, blockage of VEGF using VEGF Trap in the rhesus macaque and a different mouse model of simulated menstruation (pseudopregnancy followed by decidualisation induction and ovariectomy to trigger menses) has previously revealed inhibited neovascularisation and delayed reepithelialisation of the denuded endometrial surface during menstruation[11], consistent with our findings in these hypoxia/HIF-1 deficient models. We have previously demonstrated that VEGF mRNA and protein are increased in human endometrial tissue during menstruation and that HIF-1α is necessary for hypoxia-induced increases in *VEGF* in human endometrial epithelial cells[12]. HIF-1α has been demonstrated to directly bind to the *Vegf* promoter during menstruation in a mouse model[38]. Our data herein substantially extend these findings to provide definitive evidence that hypoxia regulates HIF-1α in vivo to coordinate timely repair of the denuded endometrial mucosal surface at menses.

Interestingly, our DMOG-treated mice showed no additional increases in repair grade over vehicle treated mice. We propose the intense physiological hypoxia observed during menstruation in our model resulted in sufficient HIF-1α stabilisation and activation in vehicle treated mice to enable efficient repair and that further HIF stabilisation with DMOG was unnecessary. This is in contrast to other tissue sites where physiological hypoxia does not occur and DMOG treatment is known to significantly increase HIF-1α protein and *Vegf* when compared to controls[39, 40].

We demonstrate that inhibition of HIF-1 with echinomycin dramatically reduces neutrophil influx during menstrual bleeding. A previous study has demonstrated reduced endometrial repair following neutrophil depletion[41], highlighting the requirement for innate cell influx for normal menstruation. Macrophage numbers were not found to be different with HIF-1 inhibition/stabilisation. This is consistent with findings in the tumour environment, where macrophage numbers were not altered by hypoxia but phenotype was affected[42]. The role of hypoxia and HIF-1 on endometrial macrophage phenotype remains to be determined.

Herein, we demonstrate an impact of HIF-1 manipulation on endometrial epithelial cells, with proliferation of luminal epithelial cells visible at an earlier time point in mice with HIF-1α stabilisation. Similar epithelial benefits were observed in a mouse model of colitis, with overexpression of HIF-1 in intestinal epithelial cells found to be protective with diminished clinical symptoms[43]. Furthermore, pharmacological stabilisation with DMOG was also found to be protective in a colitis mouse model, demonstrating an anti-apoptotic phenotype in intestinal epithelial cells that the authors suggested maintained the epithelial barrier[44]. This significant positive impact of HIF-1 on epithelial function and the prevention of pathology in an inflammatory environment are consistent with our findings in the menstrual endometrium.

Our experiments also suggest endometrial endothelial cells have optimised function with stabilisation of HIF-1α. Endometrial vessels in DMOG-treated mice were occasionally surrounded by αSMA positive cells indicative of increased maturity versus echinomycin-treated mice. This is consistent with findings in $Phd2^{+/-}$ mice (decreased HIF-1α degradation), where tumour vessel density and lumen size was unaffected but endothelial lining and maturation were normalised, resulting in improved tissue oxygenation and inhibition of metastases[45]. Our finding that silencing of HIF-1α in human endometrial epithelial cells significantly decreased endothelial cell branching further support a role for HIF-1 in vascular regulation during human menstruation. A previous study revealed differences in the vascular smooth muscle cell differentiation markers in endometrial vasculature from women with HMB and NMB[46]. Hence, HIF-1 may increase the efficiency of repair of the damaged vasculature and/or drive angiogenesis to optimise vascular function in subsequent menstrual cycles.

Finally, we reveal that DMOG stabilisation of HIF-1α in our hyperoxia model (i.e. non-hypoxic menstruation) significantly improved endometrial repair. For women with a defective hypoxic response at menstruation, this offers real promise of a non-hormonal medication to improve prolonged menstrual bleeding. Similar to effects in the mouse model of colitis[44], the actions of DMOG in the endometrium appear to be secondary to improved epithelial function, with reepithelialisation of the denuded surface a major contribution to the improved histological repair scores. Previous studies have also identified an anti-inflammatory effect of PHD inhibition[47] describing NFκB inhibition and switch to an M2 macrophage phenotype. However, we found an increase in inflammatory mediator mRNA concentrations in the mouse uterus after treatment with DMOG. These differences may be due to the origin of the inflammatory stimulus, i.e. LPS induced 'injury' versus the physiological inflammatory response of endometrial shedding (i.e. menstruation). Alternatively, the response within discrete immune cell populations may be masked in our homogenated tissue samples and anti-inflammatory effects in these populations remain undetermined.

Hydroxylase inhibitors have recently entered clinical trials for the treatment of chronic kidney disease associated anaemia and were well tolerated[48, 49]. This opens up the possibility of using these compounds for the treatment of HMB. Our findings suggest their use just prior to/during menstruation may significantly reduce the duration of bleeding. The off target effect of increased erythropoiesis may be a very desirable additional health benefit for women suffering from heavy periods, who commonly experience anaemia and occasionally require blood transfusion.

In conclusion, our data reveal an important role for hypoxia and HIF-1 in menstrual physiology. Targeting the HIF-1 pathway at menstruation has promising therapeutic potential, offering a non-hormonal, fertility preserving medical treatment option for women with prolonged HMB.

## Methods

**Human tissue collection.** Endometrial biopsies ($n = 91$) were collected with a suction curette (Pipelle, Laboratorie CCD, Paris, France) from women (median age 42 years, range 26–50) attending gynaecological out-patient departments in NHS Lothian. Written informed consent was obtained from participants and ethical approval granted from Lothian Research Ethics Committee (REC 08/S1103/38, 10/S1402/59). All women reported regular menstrual cycles (21–35 days) and no exogenous hormone exposure for 2 months prior to biopsy. Women with large fibroids (>3 cm) or endometriosis were excluded. Tissue was divided and (i) placed in RNA later, RNA stabilisation solution (Ambion (Europe) Ltd., Warrington, UK), (ii) fixed in 4% neutral buffered formalin for wax embedding. Biopsies were consistent for (i) histological dating using criteria of Noyes et al.[50], (ii) reported last menstrual period and (iii) serum P4 and estradiol concentrations at time of biopsy. Six tissue samples were excluded from analysis due to inconsistent dating and one sample excluded due to detection of hyperplasia. Classifications of participant samples are detailed in Supplementary Table 1.

**Objective measurement of MBL.** A subset of the participants ($n = 75$) agreed to collect their sanitary ware to allow objective quantification of their MBL. Women were provided with the same brand of tampon/pad (TampaxAlways) and verbal and written instructions on collection. Blood loss was measured using a modified alkaline-haematin method[51, 52]. In brief, used sanitary products were added to a measured volume of 5% sodium hydroxide. After 24 h an aliquot was removed, filtered and the optical density (OD) measured with spectrophotometry at 546 nm. This was compared to a 1 in 200 dilution of the patient's venous blood in 5% sodium hydroxide incubated for 24 h. MBL was calculated using the equation (1).

$$\text{MBL} = \frac{(\text{OD of menstrual blood solution} \times \text{total value of added sodium hydroxide})}{(\text{OD of venous blood solution} \times 200)}.$$

$$(1)$$

This method was validated using time expired whole blood applied to the same sanitary products given to participants. An MBL of >80 ml was classified as HMB and <80 ml as NMB. In total, 44 women also completed a menstrual pictogram chart, detailing their duration of bleeding.

**Mouse studies.** All experimental animal procedures were carried out with appropriate approval. The Institutional Animal Care and Research Advisory Committee of the University of Leuven granted permission to the Vesalius Research Centre, KU Leuven and UK home office approval was granted for studies at the University of Edinburgh.

**Simulated mouse model of menstruation.** Female C57BL/6JOlaHsd mice were purchased from Envigo (Hillcrest, UK). Endometrial shedding and repair was simulated in ovariectomised mice[53, 54] (Fig. 2a). In brief, 6–9-week-old female mice were ovariectomised on day 1 of the protocol to deplete endogenous steroid production. Mice then received daily subcutaneous injections of β-estradiol (E2) in peanut oil (100 ng) on days 7–9. A progesterone implant (P4) was placed subcutaneously on day 13, mice also received daily injections of E2 (5 ng) from day 13 to 15. On day 15, decidualisation of one uterine horn was induced by intracervical injection of 20 μl peanut oil using a non-surgical transfer device (ParaTechs, Lexington, KY, USA). P4-withdrawal was induced 4 days after decidualisation (day 19) by removal of the P4 implant. Mice received an intra-peritoneal injection of bromodeoxyuridine (BrdU, 100 μl) and pimonidazole (Hypoxyprobe, 60 mg/kg) 1.5 h prior to culling. Mice were culled by cervical dislocation at the time of P4-withdrawal (T0) or 8 or 24 after P4-withdrawal (T8, T24). Uteri were dissected and collected in RNA later, 4% neutral buffered formalin for paraffin embedding and snap frozen in liquid nitrogen for protein extraction. Any animal with failed decidualisation was excluded from the study. Eight replicates at each time point will give 80% power to detect as significant true mean differences of about 1.6 standard deviations between groups.

**Hyperoxic conditions.** C57BL/6 mice undergoing the menstrual protocol were randomised using Microsoft Excel to standard conditions (21% O2) or incubation in a hyperoxic chamber (75% O2, Coy laboratory Products, Michigan, USA) at the time of P4 withdrawal (T0) until sacrifice (T8 or T24). Body and uterine weights did not vary between groups (Supplementary Fig. 7a).

**Generation of HIF-1α deficient mice.** HIF-1α[+/−] ES cells were used to generate HIF-1α[+/−] mice (129/Sv × Swiss) as previously described[45, 55]. Wild-type (50% Sw 50% 129S6) females were mated with HIF-1α[+/−] (50% Sw 50% 129S6) males and offspring genotyped to identify female HIF-1α[+/−] and normal littermate controls. Mouse and decidualised uterine weights did not significantly differ between the wild-type and HIF-1α[+/−] mice (Supplementary Fig. 7b).

**Pharmacological manipulation of HIF.** C57BL/6JOlaHsd mice (Envigo) underwent the simulated menstruation protocol described above. Twenty four hours prior to P4 withdrawal (day 18) and at the time of P4 withdrawal (day 19, T0) mice were randomised to administration of an intraperitoneal injection of either echinomycin (HIF-1 inhibitor, 1 mg/kg, Bioviotica, Liestal, Switzerland), dimethyloxalylglycine (DMOG; PHD inhibitor; 8 mg, Enzo Life Sciences, Lausen, Switzerland) or vehicle control. Body weight was unaffected by drug treatment (Supplementary Fig. 7c). Decidualised uterine weight was significantly reduced 24 h after P4-withdrawal in mice treated with echinomycin (T24). No differences were observed at 8 h post withdrawal (T8) (Supplementary Fig. 7c).

**DMOG rescue.** C57BL/6J mice undergoing the simulated menstruation protocol were randomised to an intraperitoneal injection of dimethyloxalylglycine (8 mg, Enzo Life Sciences, Lausen, Switzerland) or vehicle control 24 h prior to P4-withdrawal and at T0. Mice were all placed into hyperoxic conditions (75% O2) until sacrifice at T24 (24 h post P4-withdrawal; Fig. 6a).

**Histological analysis.** Five micrometer mouse uterine sections were stained with haematoxylin and eosin (H&E) and stage of breakdown/repair graded by two masked independent observers using a previously published scoring system[41] (Supplementary Fig. 2). Mouse uterine sections were also stained for endoglin (1:50, R&D Systems, AF1320, Abingdon, UK), alpha smooth muscle actin (αSMA, 1:250, C6198, Sigma, Dorset, UK), Ly6G (BioLegend, 127601, 1:1000. London, UK), F4/80 (Bio-Rad, MCA497GA, 1:50. Oxford, UK) and carbonic anhydrase IX (Abcam, ab184006, 1:2000, Cambridge, UK). Proliferating cells were identified using anti-bromodeoxyuridine antibody (BrdU, 1:1500, Fitzgerald, Acton, MA, USA) and hypoxia detected using anti-pimonidazole antibody as per manufacturer's instructions (4.3.11.3 mouse MAb, Hypoxyprobe, Burlington, USA).

**Western blots.** In total, 15 μg of nuclear protein extract from whole human endometrial biopsies was denatured for 5 min at 90 °C. Proteins were separated on 4–12% Bis–Tris Gels (NuPAGE Novex, Invitrogen, Carlsbad, CA) and transferred onto polyvinylidene difluoride membranes (Millipore, Watford, UK). Membranes were blocked overnight before incubation with primary antibodies for 2 h at room temperature; anti-HIF1α antibody (BD Biosciences, Oxford, UK, 1:250), anti-HIF2α antibody (Novus Biologicals CO, USA, 1:1000), rabbit polyclonal to beta-actin (Abcam, Cambridge, UK 1:5000). After washing, the membrane was incubated with an appropriate secondary antibody before detection using

chemiluminescent horseradish peroxidase substrate (Immobilon Western, Millipore Corporation, MA, USA) (Supplementary Fig. 8).

In total, 80 μl whole protein extracts from whole mouse uterine biopsies were denatured and loaded to NuPAGE Novex 3–8% Tris-acetate gel. Transfer was conducted semiwet, using nitrocellulose membrane prior to blocking for 1 h. Anti-mouse HIF1α (0.5 μg/ml, AF1935, R&D Systems, Abingdon, UK) or HIF2α (1 μg/ml, AF2997 R&D Systems) was applied overnight at 4 °C. After washing, the membrane was incubated with an appropriate secondary antibody before detection using chemiluminescent horseradish peroxidase substrate (Pierce ECL Western Blotting Substrate, Life technologies, Paisley, UK). (Supplementary Fig. 8).

**Quantitative real time PCR.** Total RNA from whole uterine mouse tissue, human cells and whole human endometrial biopsies were extracted using the RNeasy Mini Kit (Qiagen Ltd., Sussex, UK) with on column DNaseI digestion according to manufacturer's instructions. RNA samples were reverse transcribed using Superscript Vilo cDNA synthesis Kit (Life Technologies). Messenger RNA transcripts were quantified relative to appropriate reference genes (human samples: 18S and ATP5B, mouse samples: ACTB and RLP13), as determined by geNorm assay (Primerdesign Ltd., Southampton, UK). Specific primers were designed using the universal probe library assay design centre and checked with BLAST (Supplementary Table 2). Reactions were performed in triplicate using ABI Prism 7900 system under standard conditions with Invitrogen 2xExpress Supermix (Life Technologies). Quantification was performed using the $2^{-\Delta\Delta Ct}$ method after normalisation against controls (a sample of liver cDNA).

**Endothelial cell branch assay.** HIF-1α was silenced in endothelial cells (Ishikawa, 99040201, Culture collections, Sailsbury, UK) using two different lentiviral ShRNA constructs (HIF-1α/shRNA1470, HIF-1α/shRNA2192), gifted by T. Cramer and described in ref. [12]. Cells were tested to exclude mycoplasma contamination. Untransfected cells and cells transfected with HIF-1α/shRNA1470, HIF-1α/shRNA2192 or HIF-1α/shRNASCR (a scrambled negative control) were covered with 1 ml of serum free DMEM media and placed in hypoxic conditions (0.5% $O_2$) for 8 h. The conditioned media (CM) was harvested. An in vitro endothelial cell branch assay was then performed[56]. In brief, 100 μl Matrigel (BD Biosciences, Bedford, MA) was added per well in a 48 well plate. Human umbilical vascular endothelial cells (HUVEC, PromoCell, Heidelberg, Germany) were seeded at a density of $2 \times 10^4$ in 200 μl EBM-2 media (Lonza, MD, USA) supplemented with GA1000 and ascorbic acid SingleQuots. The CM from (i) untransfected Ishikawa endometrial epithelial cells, (ii) HIF-1α/shRNA1470 transfected cells, (iii) HIF-1α/shRNA2192 transfected cells, (iv) HIF-1α/shRNASCR transfected cells (v) HIF-1α/shRNA1470 transfected cells plus 300 ng ml$^{-1}$ rhVEGF protein or (vi) HIF-1α/shRNA2192 transfected cells plus 300 ng ml$^{-1}$ rhVEGF protein. Assays were carried out in triplicate and with CM from three separate experiments. Endothelial cell branching was captured in the same position in each well after 8 h using an inverted Axiovert microscope. Branch points of the formed tubes were counted by an observer masked to the sample origin and an average of the replicates determined after unmasking.

**Statistical analysis.** Analysis was carried out using GraphPad Prism software (San Diego, CA). Analysis of two groups (i.e. histological breakdown/repair scores, densitometry, duration of bleeding) utilised unpaired t tests where the distribution of data was normal and Mann–Whitney test where data were not normally distributed. For comparison of data sets with two variables (i.e. human data: MBL and stage of cycle, mouse data: treatment groups and time) a two way analysis of variance was used with Bonferroni's/Tukey's multiple comparisons test. Endothelial cell branch point assay was analysed by paired one-way analysis of variance with Tukey's multiple comparisons test. A value of $P < 0.05$ was considered statistically significant.

**Data availability.** All relevant data are available from the authors upon reasonable request.

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

## Acknowledgements

We thank M. Nicol, K. Walker, D. Wilson, R, Murgai, H. Murray, L. Boswell, A. Carton, A. Manderveld, K. Brepoels, P. Vanwesemael, E. Van Dyck, R. Martinez-Aguilar and A Dearbey for technical support. We are grateful to A. Williams for scoring uterine histology and P. Brown for construction of lentiviral vectors. Thanks to R. Grant and S. Milne for help with figure and manuscript preparation and C. Murray and S. McPherson for patient recruitment. We are grateful to S. Walmsley and K. Chapman for critical appraisal of the manuscript. This work was supported by the Wellcome Trust (100646/Z/12/Z and ISSF fund J22738), Academy of Medical Sciences (AMS-SGCL13), Wellbeing of Women (RG1820) and the Medical Research Council (G0600048, G0500047) and was carried out in the Medical Research Council Centre for Reproductive Health, which is funded by MRC Centre grants G1002033 and MR/N022556/1.

## Author contributions

J.A.M., H.O.D.C., N.H., and P.T.K.S. conceived and designed the study. J.A.M. performed the majority of the experiments, analysed the data and prepared the manuscript. A.A.M. performed experiments and contributed to study design. P.C. contributed to the design and interpretation of the study and provided HIF-1α heterozygote mice. All authors provided critique and approved the submission.

## Additional information

**Competing interests:** J.A.M., A.A.M., P.T.K.S., N.H. and P.C. declare no conflicts of interest. H.O.D.C. has received clinical research support for laboratory consumables and staff from Bayer Ag and provides consultancy advice (but with no personal remuneration) for Bayer Ag, PregLem SA, Gideon Richter, Vifor Pharma UK Ltd., AbbVie Inc. and Myovant Sciences.

