## [Peer Review File · Nature Communications]

Reviewers' comments:

Reviewer #1 (Remarks to the Author):

It has been well established that in the absence of implantation and the subsequent fall in progesterone leads to menstrual bleeding of the endometrium. The results of the classic study of menstruation in rhesus monkeys by Markee, conducted several decades back, still form the basis of our understanding of the mechanisms of menstruation, which showed that vasoconstriction of the spiral arteries leads to the bleeding and sloughing of the upper endometrial zones. Despite the clinical implications of excessive bleeding, the understanding of the local factors and mechanisms of menstruation is not yet fully at hand. Although spiral artery constriction and the resulting hypoxia in the upper endometrial zones are believed to be the primary stimulus for induction of menstruation, the expression and roles of hypoxia inducible factor (HIF-1 α) in the endometrium remain controversial, with conflicting results from different laboratories using human tissue samples and animal models. This study by Maybin et al. demonstrates that transient hypoxia and the resulting stabilization of hypoxia inducible factor (HIF-1 α) is essential for normal menstruation and the postmenstrual repair of endometrium, and impaired HIF-1 α stabilization during menstruation is associated with prolonged menstrual bleeding.

The authors have used well controlled normal human endometrial tissue samples and endometrial samples from women with heavy menstrual bleeding, and a well-established mouse model of endometrial breakdown and repair. In the mouse model, they induced both hypoxia and hyperoxia to test their hypotheses and used tissue localization of pimonidazole to identify hypoxic cells. This is a well-designed study on the role of HIF-1 α in the endometrium. They have shown that endometrial HIF-1 α is posttranscriptionally regulated, likely by stabilization of the protein. Women with HMB had reduced endometrial HIF-1 α and downstream target genes and prolonged menstrual bleeding likely due to delayed endometrial repair. In the mouse model, active bleeding was observed in hypoxic tissues and prevention of menstrual hypoxia resulted in decreased HIF-1 α and delayed repair of the endometrium. They have also demonstrated that administration of a HIF stabilizer at the time of menstruation significantly improved endometrial healing and repair. This is particularly important as hydroxylase inhibitors have recently entered clinical trials for the treatment of chronic kidney diseases associated anemia, and thus, has potential use for the treatment of HMB. In sum, these data together suggest an important role for hypoxia and HIF-1 α in menstrual induction cascade, and targeting the HIF-1 pathway at menstruation has promising therapeutic potential. However, the authors and other laboratories have extensively studied the role of hypoxia in leukocyte recruitment and their role in menstruation physiology. The readers would benefit if the authors could show the changes, if any, in endometrial leukocyte populations in response to HIF-1 stabilization. The manuscript is well written and provides new insights on the role of HIF-1 α stabilization in menstrual physiology, and is suitable for publication in Nature Communications.

Reviewer #2 (Remarks to the Author):

In the present study, Maybin et al investigate hypoxia and HIF in menstruation control. They use human material and a mouse model they have establish. This study is very interesting and has a high potential for therapeutic intervention in women suffereing from HMB. While most of the data is sound and consistent with the authors explanations and claims, some parts are not clearly supported by the data. In addition, minor edits in the labelling of some of the graphs is required for the non-expert to follow.

Specific points:

Major inconsistency is the role for VEGF in all the models. There was no difference in the mouse

genotypes but the phenotype was very strong, hence it is most likely that another HIF target is lacking in the HIF hets to generate the phenotype. Although, it would not fall in the remit of this study the authors have to at least discuss these inconsistencies. Similarly, by reducing VEGF in the branching assay, this should lead to less branches much like HIF knockdown, but again no difference.

The role of HIF2alpha has not been investigated properly. Especially, in disease conditions, it could be that loss of HIF-1 is leading to increased HIF2 activity. This has been seen in many systems. So using the HETs or HMB samples, HIF2a levels should be analysed.

The data on figure 6 is very interesting and should be compared to the levels of normoxia to establish how much of a rescue does DMOG produce.

Minor points

Figure 1 loading if very uneven

Figure 1F, x-axis labeled has loss (ml) but there is not numbers there

Figure 3C, is the label from 3D applicable to 3C? Not clear

Figure 5D and E need quantification, not clear from the qualitative data.

Response to reviewers

Reviewer #1

1. The readers would benefit if the authors could show the changes, if any, in endometrial leukocyte populations in response to HIF-1 stabilization.

Response: We thank our Reviewer for their helpful comments. We agree that the effects of HIF-1 on the leukocyte population would be of interest to the reader. Therefore, in response to the request of our Reviewer we have examined neutrophil and macrophage cells in tissue from mice treated with echinomycin (HIF-1 inhibition), vehicle or DMOG (HIF- α stabilisation). We have added these data to our manuscript (Figure 5D, Supplementary Fig. 4B, results Page 12, paragraph 2, Discussion Page 19, paragraph 1, Methods Page 25).

Reviewer #2

1. Major inconsistency is the role for VEGF in all the models. There was no difference in the mouse genotypes but the phenotype was very strong, hence it is most likely that another HIF target is lacking in the HIF hets to generate the phenotype. Although, it would not fall in the remit of this study the authors have to at least discuss these inconsistencies. Similarly, by reducing VEGF in the branching assay, this should lead to less branches much like HIF knockdown, but again no difference.

Response: We thank our reviewer for the comments regarding the *Vegf* data in our models. We can confirm that *Vegf* was significantly decreased with pharmacological inhibition of HIF-1 (Figure 5C) at the time of endometrial hypoxia (T8: active bleeding). The reviewer correctly points out that there was no significant difference in *Vegf* between HIF-1 α heterozygotes and WT mice or between hypoxic vs non-hypoxic menstruation, although there was a trend towards decreased uterine *Vegf* and other downstream targets of HIF-1 in non-hypoxic menses (Figure 3E and Supplementary Figure 2). We propose this lack of detection of a significant decrease in *Vegf* in our HIF-1 α heterozygote model may be due to two factors: (1) mice are heterozygous for HIF-1 α , rather than having complete knockout, which is embryonically lethal, meaning differences are more subtle. (2) Menstrual physiological hypoxia is localised to the denuded endometrial surface and mRNA was measured in whole uterine samples. This means localised, clinically significant changes may not be detected. To address this we have added carbonic anhydrase IX (as an endogenous sensor of HIF activity) immunohistochemistry data to Figure 4 (Figure 4D) and have added text to the Results (Page 11, paragraph 1), Discussion (Page 17, paragraph 2) and Methods (Page 25). We hope this confirms to the reviewer that our findings in the mouse models are robust.

Regarding our human cell branching assay. Culture supernatant from HIF-1 α silenced endometrial cells resulted in significantly reduced endothelial branching. Addition of VEGF to this HIF-1 α silenced culture supernatant restored endothelial cell branching. We

apologise that this was not clear and have amended our manuscript to ensure the reader does not think that VEGF alone was reduced (Page 14, paragraph 1).

2. The role of HIF2alpha has not been investigated properly. Especially, in disease conditions, it could be that loss of HIF-1 is leading to increased HIF2 activity. This has been seen in many systems. So using the HETs or HMB samples, HIF2a levels should be analysed.

Response: We thank our Reviewer for raising this important point. Unfortunately the nuclear extracts from human endometrial biopsies, which have been carefully categorised for stage of the menstrual cycle and have matched objective blood loss measurements, are very difficult to obtain and yield very small amounts of tissue. Therefore, generation of HIF-2a Westerns is not possible in these human samples. However, we have performed the requested Western blots for HIF-2a in our mouse HIF-1a versus wildtype samples. These important data have been added to our manuscript, displaying no significant differences in HIF-2 α protein in wild type and HIF-1 α heterozygous mice at T8 (Supplementary Figure 3B, Results Page 10, paragraph 2, methods Page 26). This lack of compensation is consistent with the phenotype we observed in HIF-1 α heterozygous mice.

3. The data on figure 6 is very interesting and should be compared to the levels of normoxia to establish how much of a rescue does DMOG produce.

Response: Almost 60% of mice incubated in normoxic conditions (hypoxic menses) reached stage 5 of endometrial repair at T24 (Figure 3D). In comparison, almost 80% of mice in hyperoxic conditions treated with DMOG reached stage 5. Our normoxia versus hyperoxia experiment was carried out prior to the rescue experiment with DMOG. To limit animal use, in accordance with the UK Animals (Scientific Procedures) Act 1986, we did not repeat the normoxic incubation in our rescue experiment as this had already been carried out. There was a significant improvement in endometrial repair in mice treated with DMOG during non-hypoxic menstruation (which we propose is analogous to the defects in women with HMB), indicating a clinically significant improvement.

Minor points

Figure 1 loading is very uneven

Response: We acknowledge that the B-actin blot is uneven. Unfortunately these nuclear extracts from human endometrial biopsies that have been carefully categorised for stage of cycle with matched objective blood loss measurements are very difficult to obtain and yield small amounts of tissue. Therefore, further repetition of these Western blots is not possible. As loading differences are accounted for in our densitometry analysis (Figure 1B/C/D) and the data are consistent with downstream target gene expression (Figure 1E) and findings in our mouse models, we hope the reviewer is reassured of their accuracy.

Figure 1F, x-axis label has blood loss (ml) but there is not numbers there

Response: x-axis label amended in **Figure 1F**.

Figure 3C, is the label from 3D applicable to 3C? Not clear

Response: Figure legend amended (**Page 39**).

Figure 5D and E need quantification, not clear from the qualitative data.

Response: We understand this request for quantification but as this is immunohistochemical data any analysis would be semi-quantitative at best. Our concern is that quantification may detract from the spatial differences seen between our treatment groups, e.g. intense BrdU staining in the reepithelialising luminal epithelium, and give a biased comparison between groups. On balance, we feel that a panel of images is more informative than an attempt at quantification. To aid clarity we have removed the T0 time-point, where no differences were observed between treatment groups and have added arrows to areas of interest at T24 (**now Figure 5E**). Regarding vascular staining, we observed differences in morphology between treatment groups, rather than a quantitative defect. Therefore, we have chosen to display representative images in Figure 5F.

REVIEWERS' COMMENTS:

Reviewer #1 (Remarks to the Author):

The manuscript has been improved significantly since the first submission. The authors have added substantial new data, including changes in neutrophil and macrophage infiltration in response to various treatments, and clarified all the questions raised by the editors/reviewers.

Reviewer #2 (Remarks to the Author):

The authors have addressed all of my previous concerns

REVIEWERS' COMMENTS:

Reviewer #1 (Remarks to the Author):

The manuscript has been improved significantly since the first submission. The authors have added substantial new data, including changes in neutrophil and macrophage infiltration in response to various treatments, and clarified all the questions raised by the editors/reviewers.

Response: Many thanks for your further review of our manuscript and positive comments.

Reviewer #2 (Remarks to the Author):

The authors have addressed all of my previous concerns

Response: Many thanks for your further review of our manuscript and positive comments.